# In-Vitro Investigation of Marginal Adaptation and Fracture Resistance of Resin Matrix Ceramic Endo-Crown Restorations

**DOI:** 10.3390/ma16052059

**Published:** 2023-03-02

**Authors:** Burak Mertsöz, Salim Ongun, Mutahhar Ulusoy

**Affiliations:** 1Department of Prosthodontics, Faculty of Dentistry, Near East University, Mersin 10, Lefkosa 99040, Turkey; 2Department of Prosthodontics, Faculty of Dentistry, Final International University, Nicosia 99010, Turkey

**Keywords:** endo-crown restoration, fracture resistance, marginal fit, resin matrix ceramics

## Abstract

The aim of this study was to evaluate the fracture resistance and marginal adaptation of endo-crown restorations produced from different resin-matrix ceramics (RMS) and the effects of these materials on marginal adaptation and fracture resistance. Three frasaco models were used by preparing (first) premolar teeth in three different margin preparations: butt-joint, heavy chamfer and shoulder. Each group was further divided into four subgroups according to the type of restorative material used: Ambarino High Class (AHC), Voco Grandio (VG), Brilliant Crios (BC) and Shofu (S) (*n* = 30). Master models were obtained using an extraoral scanner and fabricated with a milling machine. Marginal gap evaluation was performed with a silicon replica technique using a stereomicroscope. Replicas of the models (*n* = 120) were produced with epoxy resin. The fracture resistance of the restorations was recorded using a universal testing machine. The data were statistically analyzed using two-way ANOVA, and a t-test was applied for each group. Tukey’s post-hoc test was performed to compare significant differences (*p* ≤ 0.05). The highest marginal gap was observed in VG, and the best marginal adaptation and the highest fracture resistance were found in BC. The lowest fracture resistance in Butt-joint preparation design was found in S. In addition, the lowest fracture resistance value in the heavy chamfer preparation design was found in AHC. The heavy shoulder preparation design displayed the highest fracture resistance values for all materials.

## 1. Introduction

Treatment options for the restoration of endodontically treated teeth with extensive crown damage are increasing with the development of new materials. One of these treatment options is endo-crown restoration, indicated in teeth with short clinical crown length, insufficient interocclusal distance, calcified canals or thin-rooted teeth, which has been reported to be a promising treatment [1,2]. An endo-crown restoration can be defined as a one-piece restoration that replaces the crown macro-mechanically retained with an extension into the pulp chamber [3], thus making it a more suitable treatment option than onlay, overlay and post-core restorations that uses the whole pulp chamber as a retentive resource [1].

Several parameters can affect the mechanical behavior of an endo-crown restoration, including the usage of different restorative materials, the finish line design and the depth of the pulp chamber [4].

Recently introduced resin matrix ceramics (RMCs) consist of materials with a highly infiltrated organic matrix with ceramic particles and have a modulus of elasticity closer to dentin than conventional ceramics. RMCs merge the advantageous characteristics of dental ceramics and composite resins and thereby provide the following benefits: easy and fast machinability, can be repaired intraorally, less chipping, better fracture resistance, acceptable wear resistance, low abrasive effect on opposing teeth, near-to-ideal bond strength, polishability and no need for firing [5,6,7,8].

The morphologic fit between the edges of the restoration and preparation is referred to as the marginal adaptation [9,10]. The adaptation of endo-crowns, both marginal and internal, plays an important role in their clinical outcome [1]. In cases where the marginal adaptation of the restorations is not fit, the dissolution of the cement over time can cause endodontic and periodontal inflammation as a result of bacterial microleakage causing failure of the prosthesis [11,12]. In addition to material selection, ferrule and finish line preparation of teeth restored with endo-crown restorations have been reported to provide greater fracture resistance [13]. It also has been reported that building the foundation of the pulp chamber floor does not improve either fracture resistance or marginal adaptation.

The first null hypothesis of this study was that different margin designs will not affect the fracture resistance and marginal adaptation of endo-crown restorations. The second null hypothesis was that the type of current RMC materials will not affect the fracture resistance and marginal adaptation of endo-crown restorations.

This study aims to evaluate the fracture resistance strength and marginal adaptation of the endo-crown restorations produced from different types of RMC materials and to evaluate the effects of these RMC materials and different margin configurations on marginal adaptation and fracture resistance strength.

## 2. Materials and Methods

### 2.1. Specimen Preparation

In this study, 3 frasaco teeth ANA-4 ZPUR (for preclinical training in root canal treatment, ivory crown with red pulp wall and rooted, suitable for X-ray control) were adapted in the maxillary left second premolar teeth of a phantom model (ANA-4 Frasaco GmbH, Tettnang, Germany). The three models were determined as the main models. Diamond-coated stainless steel burs (Hager and Meisinger GmbH, Berlin, Germany) were used to prepare the pulp chamber and finish line of these master models. The dimensions of the master models of endo-crown preparation were 2 mm occlusal reduction, a pulp chamber depth of 4 mm and a width of 6 mm and an 8° angle of artificial pulp chamber. Then, 3 main models were subjected to 3 different finish line preparations: butt-joint, heavy chamfer and heavy shoulder. The margin design of the heavy chamfer and heavy shoulder were prepared to be equal all around and 1 mm wide.

### 2.2. Teeth Preparation for Endo-Crowns

The 3D scanning process of the main model with 3 different finish line preparations was performed with an extraoral scanner (Sirona inEOS X5, Dentsply Sirona, York, PA, USA), and the obtained digital data were transferred to computer-aided design (CAD) software (inLab SW 16.1; Dentsply Sirona). A schematic illustration of endo-crown preparation and study design is presented in Figure 1.

The main model’s negative impressions were taken with A-type silicone (Elite HD+ Maxi Putty Soft-Fast Set Vinylpolysiloxane, Zhermack, Italy). Epoxy resin (Epoxy Resin EPOART Ultra Transparent, Polisan, Kocaeli, Turkey) was injected into the negative impression taken with a surgical syringe, and the main model was duplicated. The main model duplication process was carried out for 40 repetitions for the 3 main models. A total of 120 master model replicas were produced.

From 4 different RMC discs (Ambarino High Class (*n* = 30), Voco Grandio (*n* = 30), Brilliant Crios (*n* = 30), Shofu (*n* = 30)) were investigated, and the discs used in this study are presented in Table 1. All endo-crown restorations were milled with a 5-axis CAD-CAM machine (Sirona inLab MC X5, Dentsply Sirona, PA, USA), and the specimens were fabricated in 3 finish line types. Prior to cementation to master model replicas, the buccal, lingual, mesial and distal marginal adaptations of the endo-crown designs were examined with the aid of a stereomicroscope (Olympus SZ61TR, Olympus Corporation, Shinjuku, Tokyo, Japan) at 40× magnification.

### 2.3. Evaluation of Marginal Adaptation before Cementation

In this study, a stereomicroscope was used for the evaluation of marginal adaptation. The silicone replica technique was used for easier monitoring of marginal measurements. Since we did not examine the internal adaptation of the specimens in the study we performed and in order not to damage the specimens at this stage by subjecting them to fracture resistance later, we did not apply the scalpel sectioning process; instead, we only examined the marginal adaptation. In the silicone replica technique, light-body silicone material (Elite HD + Lightbody Fast Setting, Zhermack, Italy) is mixed and applied into endo-crown restorations with the help of a mixing gun (Applyfix 4, Kettenbach GmbH & Co KG, Eschenburg, Germany) to mimic the cement gap and temporarily cement the restoration to the epoxy resin framework. The reason for this procedure is that it is easier to examine the marginal gap under the stereomicroscope at 40× magnification. Marginal gap measurements were taken for each shot (four equidistant landmarks along the cervical circumference for each surface of the specimen (mesial, buccal, distal and palatal). Each measurement was repeated five times. The images were calibrated each time to ensure the same distance. The average of the micrometer values was calculated by using the measurement software available in the stereomicroscope to calculate the range values at these measurement points (Figure 2).

### 2.4. Cementation and Fracture-Strength Test

Single Bond Universal was applied to the pulp chambers of the replicas (Single Bond Universal, 3M ESPE, St. Paul, MN, USA). A halogen light source (LED-B, Woodpecker, 1000–1700 mw/cm^2^) was applied to the pulp chamber for 10 s. Cementation of the endo-crowns was performed by using dual cure self-adhesive resin cement (Nova Resin, IMICRYL, Konya, Turkey) under finger pressure for 60 s. The polymerization process was applied to the endo-crown from 5 surfaces (buccal, lingual, mesial, distal, occlusal) with a halogen light source for 20 s each. In order to compare the fracture resistance, the endo-crowns were divided into 4 groups according to the RMC material from which they were produced. These 4 groups were further divided into 3 subgroups according to their margin design. Fracture resistance was applied to a total of 12 groups using a universal test machine (EZ50 Universal Test Machine 50 kN, Ametek Lloyd Instruments Ltd., West Sussex, UK). The load was implemented through a steel rod with a 4-mm rounded tip, and the tip was positioned at the central fossa of the endo-crown. Static loading was performed at a crosshead speed of 1 mm/min until fracture occurred, and the fracture loads were recorded in Newton.

### 2.5. Statistical Analysis

Power analysis was performed to calculate the required sample size using an analysis software (G*Power, Version 3.1.9.3. for Mac). The sample size of each group was calculated to be 10 with 80% power and 95% confidence level at α = 0.05. Ten specimens were prepared for each test group to ensure the targeted statistical power.

Statistical analysis of all data from both marginal adaptation and fracture resistance tests was performed using a statistical software program (SPSS, IBM Statistics 23.0, Chicago, IL, USA). Marginal adaptation and fracture resistance values were separately analyzed with two-way analysis of variance (ANOVA). T-tests were applied for each group, and Tukey’s post-hoc test was performed to compare significant differences between groups. Statistical significance was accepted as *p* ≤ 0.05.

## 3. Results

The mean values of the marginal gap for all groups are summarized in Table 2. The results of the two-way ANOVA parametric test revealed a statistically significant difference between the groups (*p* ≤ 0.05). It was observed that the type of preparation did not affect the marginal adaptation (*p* = 0.00). It was concluded that the material used (*p* = 0.00) and the interaction of the material with the preparation type (*p* = 0.02) significantly affected the marginal adaptation. While the highest marginal gap values were observed in Voco Grandio (36.53 ± 12.01 μm), the best marginal adaptation was found in Brilliant Crios (27.58 ± 4.34 μm) compared to all other groups.

The mean values and standard deviations of fracture resistance for different CAD-CAM blocks of endo-crown restorations are presented in Table 3. The results of the two-way ANOVA parametric test revealed a statistically significant difference between the groups (*p* ≤ 0.05). A significant difference was observed in the fracture resistance of the materials, the finish line types and the interaction between the material and the finish line (*p* = 0.00). Brilliant Crios (828.10 N) has the highest fracture resistance value and Shofu (586.18 N) has the lowest fracture resistance. The lowest fracture resistance in butt-joint preparation design was found in Shofu (357.73 N). In addition, the lowest fracture resistance value in heavy chamfer preparation design was found in Ambarino High Class (517.46 N). The heavy shoulder preparation design displayed the highest fracture resistance values in all materials.

## 4. Discussion

The first null hypothesis of the study—that different margin designs will not affect the fracture resistance—was rejected. However, marginal adaptation of endo-crown restorations was not affected, thereby confirming the hypothesis. The second null hypothesis—that the type of current RMC materials will not affect the fracture resistance and marginal adaptation of endo-crown restorations—was rejected completely. 

The clinically difficult restorations of endodontically treated teeth with extensive coronal destruction require careful planning [14]. Therefore, dentists need to decide on the most effective and best treatment option to extend the duration of the clinical prognosis of such teeth.

Endo-crown restorations are advantageous in this aspect in order to maintain the integrity of teeth, which are not ideal for single-crown or post-core restorations. Endo-crowns take advantage of contemporary developments in ceramic CAD/CAM technologies and various types of resin cement [15].

Stress levels are lower in endo-crown teeth than in full crown restorations [16]. In a study comparing endo-crowns with conventional crown restorations, it was reported that endo-crowns exhibited better tooth fracture resistance and also reported that the fracture resistance of endo-crowns was greater than that of intact premolars. In addition, the endo-crown design could restore the structural integrity and the strength of an endodontically treated and severely decayed tooth [17]. 

Sedrez-Porto et al. compared endo-crown restorations with conventional treatments using post-core, direct composite resin or inlay/onlay restorations in terms of survival time and fracture strength. According to the results of the study, it was reported that endo-crowns may perform similarly to or better than conventional restorations [18]. Belleflamme et al. reported that endo-crowns are a reliable form of treatment to restore severely damaged teeth, even in the presence of occlusal risk factors such as extensive coronal hard tissue loss, bruxism or incompatible occlusal relationships [19]. 

In an in vitro study by Biacchi and Basting (2012), the fracture resistance of endo-crown restorations and glass fiber post restorations were compared. It was reported that endo-crown restorations showed higher fracture strengths than glass fiber restorations. Post-core restorations are contraindicated, especially in some cases where the crown has extensive damage, the interproximal space is limited, ceramic thickness is insufficient and the root canals are very thin and excessively curved. In these cases, endo-crown restorations may be preferred [1].

Amongst the many materials being used for endo-crown restorations, four types of different resin matrix ceramics were used in this study. Compared to other ceramic materials, these materials have higher flexural strength and moduli of elasticity that more closely resemble the properties of dentin [6,20]. They can be milled more easily than glass ceramics and polycrystalline ceramics. In addition, the ease of repair with composite resin provides an advantage over other ceramics [5]. This provides the advantage of faster milling with less chipping in the margin area and less wear of the milling cutters [6,7]. Hassouneh et al. concluded that endo-crowns obtained from the CAD-CAM resin matrix ceramic blocks are a reliable option for restoring endodontically treated premolars [20]. When analyzing the fracture resistance of teeth restored with resin matrix ceramics compared to feldspathic all ceramic restorations, Attia et al. found no significant difference in their study [21], while Lise et al. reported a better result when using resin matrix ceramic restorations [22]. In addition, since the fracture resistance of all materials is higher than the force values obtained during chewing in the premolar region, they suggest their use in the restoration of endodontically treated premolars, but the evaluation of other mechanical properties, such as wear resistance of the materials, in future studies will provide more detailed information.

In our study using four different RMC materials, it was found that Brilliant Crios had the highest fracture resistance value (828.10 ± 223.87 N) and Shofu had the lowest fracture resistance value (586.18 ± 216.92 N). In a study conducted by Acar and Kalyoncuoğlu, similar results were obtained in endo-crown restorations, where the highest fracture resistance was observed in Brilliant Crios (2072.77 ± 454.65 N) and the lowest fracture resistance was observed in Shofu (1068.36 ± 214.91 N) [23]. The reason for this force difference between the materials in the two studies is thought to be that the surface area of the molar tooth is higher than the premolar tooth and the substructure can therefore absorb more force. The conventional viewpoint is that the mechanical properties, such as the flexural strength, the modulus of elasticity, the wear resistance or the fracture resistance, increase with filler content [24]. Compressive strengths of resin matrix ceramics can be improved by loading nano-filler particles of smaller diameter. Resin matrix ceramics have the potential to improve fracture resistance by utilizing smaller nano-filler particles [25]. Koenig et al. showed the largest filler particles were identified in Shofu Block HC, Voco Grandio and followed by Brilliant Crios [24]. Based on this information, Brilliant Crios shows the highest fracture resistance.

In another study, the fracture resistance of Voco Grandio and Shofu materials were compared and Voco Grandio can withstand pressure over 2500 N, while fractures were observed at Shofu Block at 1500 N values [26], which are similar to our study.

Kassem et al. designed endo-crown restorations with composite-containing material and stated that this material corresponds to Ambarino High-Class material. The fracture resistance of this material was found to be 2420 N. In this study, the butt-joint finish line type was used, and it was observed that the fracture resistance of this material was higher than the Ambarino High-Class material [27]. Considering the results of our study, Brilliant Crios was found to have the highest fracture resistance among the RMC materials. It was observed that the fracture resistance of the materials decreased when the finish line of the materials was designed as the butt-joint finish line, and only in the Ambarino High-Class material the butt-joint finish line preparation increases the fracture resistance of the endo-crown.

In our study, the difference in finish line preparation in the substructure design of the endo-crown restoration did not affect the marginal adaptation; however, the finish line type was found to affect the resistance of the material. Taha et al. produced endo-crown restorations using CAD-CAM and compared the fracture resistance of these restorations with butt-joint and shoulder finish line preparations [28]. The fracture resistance of an endo-crown with butt-joint preparation was found to be the lowest, while the fracture resistance of an endo-crown with shoulder finish line preparation was found to be the highest, which was similar to our study. In another study conducted by Bamajboor and Dudley, no significant difference in fracture resistance and marginal adaptation was found in restorations with butt-joint and shoulder finish line preparation [29]. 

Marginal adaptation studies have revealed quite different values ranging from 7.5 µm to 206.3 µm. This variation can be attributed to several factors, including the techniques being used to assess adaptation, the materials used, the type of preparation, the impression materials and methods, the type of cement and cementation process and the casting, modeling and processing of ceramics in the laboratory. Cementation techniques such as uncontrolled finger pressure or overfilling the crown with cement can cause an uneven flow of cement, with a thick film on one axial wall and a thin film on the opposite wall [11].

The most common methods used to measure marginal adaptation are the direct-vision technique, the cross-sectioning technique and the silicon replica technique [11]. In our study, the silicone replica technique was used to evaluate marginal adaptation. This technique is a non-invasive, inexpensive procedure that allows accurate, reproducible measurements to be made at various points, allowing both marginal and internal adaptation of restorations to be examined [30,31]. It is also a preferred technique for in vivo evaluation of indirect prosthetic restorations. Although the finger pressure applied in this technique cannot be standardized, it has been reported in the literature that the insertion force does not have a significant effect on the gap between restorations [32]. However, it also has some limitations, such as difficulties in defining the edges of the crown and the finish line, damage to the elastomeric film layer during removal from the restoration and an inability to obtain accurate cross-sections [11]. 

In the present study, the absence of the wear resistance, thermal and chewing cycle stages can be considered as a limitation, and further studies investigating and comparing these factors are needed.

## 5. Conclusions

Within the limitations of the study, the following conclusions can be reached:In endo-crown restorations, different margin preparations do not cause a change in the marginal adaptation. The type of material used affects the marginal adaptation.Except for the Ambarino High Class material, the lowest fracture resistance was observed in the butt-joint margin preparation in the other three materials, while the lowest fracture resistance was observed in the heavy chamfer margin type in the Ambarino High Class material. The marginal design to be applied in restorations using these materials should be the heavy shoulder margin design.

As a margin configuration, it is recommended that the heavy shoulder finish line type is used as an ideal margin type, and the butt-joint margin type should be avoided in order to ensure the fracture strength of the restoration.

## Figures and Tables

**Figure 1 materials-16-02059-f001:**
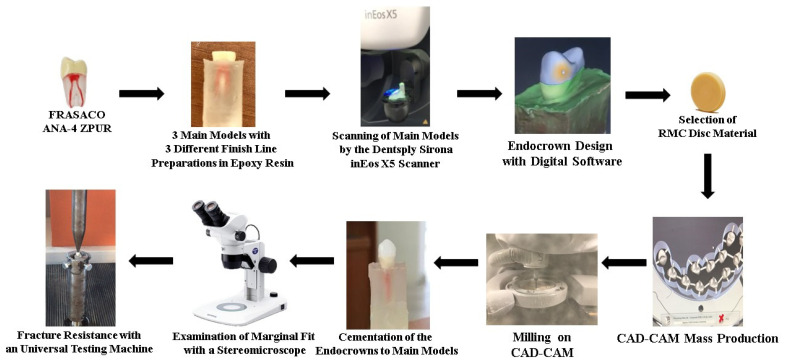
A schematic illustration of endo-crown preparation and study design.

**Figure 2 materials-16-02059-f002:**
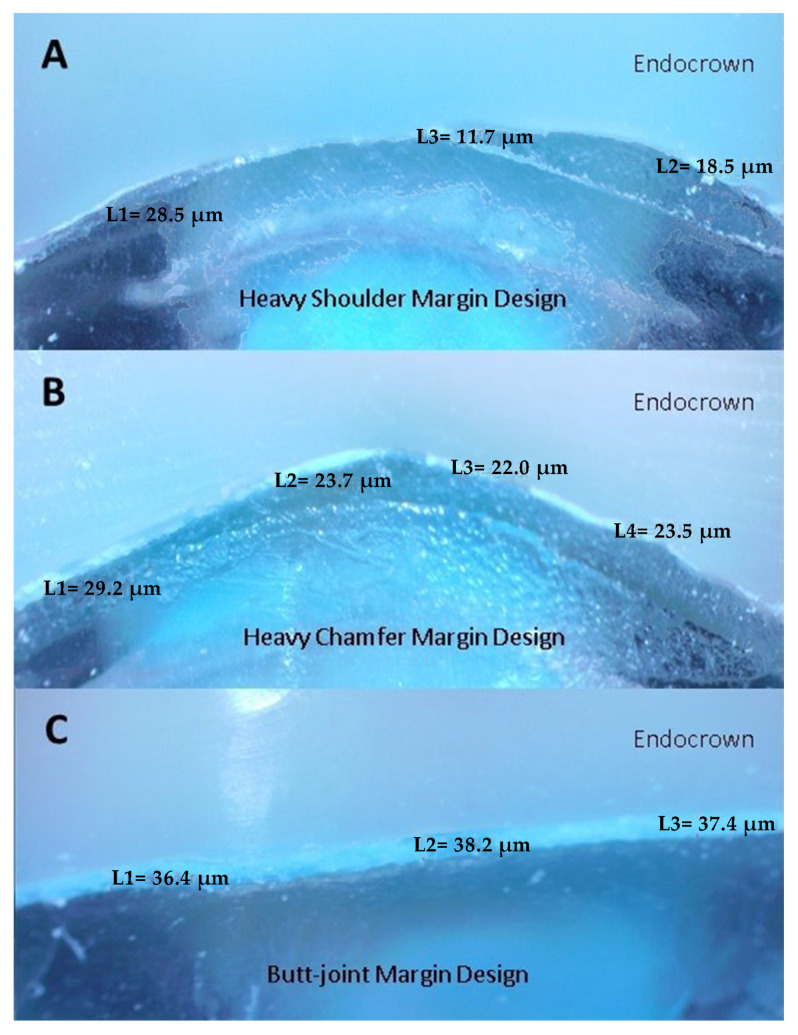
Example of marginal gap from the distal surface. (**A**) Heavy Shoulder Margin Design (**B**) Heavy Chamfer Margin Design (**C**) Butt-joint Margin Design.

**Table 1 materials-16-02059-t001:** Margin design, product, shapes, shades, ref, lot numbers and the composition of the CAD-CAM materials used in this study.

Material	Endo-Crown Preparation	Product	Shapes	Shades	Ref.	Lot	Composition
NANO CERAMIC HYBRID CAD/CAM MATERIAL	Butt-Joint	BRILLIANT Crios	14 × 98.5 (mm)	A2 LT	60022900	J69904	Organic part: cross-linked methacrylates(Bis-GMA, Bis-EMA, TEGDMA) Inorganic part: 70.7 wt% barium glass and amorphous silica
	Heavy Chamfer						
	Heavy Shoulder						
	Butt-Joint	VOCO Grandio	15 × 98.4 (mm)	A2 HT	6058	2122709	Organic part: 14% UDMA+ DMA Inorganic part: 86 wt% nanohybrid fillers
	Heavy Chamfer						
	Heavy Shoulder						
	Butt-Joint	AMBARINO High-Class	15 × 98.5 (mm)	A2	900200	121020	Organic part: 30 wt% highly cross-linked polymer blends (Bis-GMA, UDMA and BUDMA) Inorganic part: 70 wt% ceramic-like inorganic silicate glass filler particles (0.2–10 µm, average 0.8 µm)
	Heavy Chamfer						
	Heavy Shoulder						
	Butt-Joint	SHOFU	14 × 98.5 (mm)	A2 LT	2189S	121701	Filler composition: 61%, İncludes zirconium silicate, silicon dioxide, UDMA, TEGDMA
	Heavy Chamfer						
	Heavy Shoulder						

**Table 2 materials-16-02059-t002:** Values and standard deviations of marginal gap for test groups (μm).

	Brilliant Crios	Voco Grandio	Ambarino High Class	Shofu	Mean
Butt-Joint	27.47 ± 4.95 ^A,a^	30.31 ± 4.36 ^A,a^	35.44 ± 8.08 ^A,a^	32.41 ± 12 ^A,a^	31.41 ± 8.19 ^A^
Heavy Chamfer	29.89 ± 3.39 ^A,a^	39.36 ± 3.84 ^B,b^	30.76 ± 1.98 ^A,ab^	38.05 ± 2.65 ^A,ab^	34.52 ± 5.19 ^A^
Heavy Shoulder	25.39 ± 3.66 ^A,a^	39.93 ± 19.15 ^B,b^	27.42 ± 4.67 ^A,a^	30.42 ± 7.07 ^A,a^	30.79 ± 11.67 ^A^
Mean	27.58 ± 4.34 ^a^	36.53 ± 12.01 ^b^	31.2 ± 6.28 ^ab^	33.63 ± 8.56 ^b^	32.24 ± 8.84

Note: Different capital letters indicate differences in same column; different lowercase letters indicate differences in same row for each ceramic type.

**Table 3 materials-16-02059-t003:** Mean values and standard deviations of fracture resistance in Newtons (N) for test groups.

	Brilliant Crios	Voco Grandio	Ambarino High Class	Shofu	Mean
Butt-Joint	688.83 ± 185.88 ^A,b^	490.29 ± 27.93 ^A,ab^	711.14 ± 169.58 ^AB,b^	357.73 ± 154.51 ^A,a^	561.99 ± 206.17 ^A^
Heavy Chamfer	845.12 ± 168.86 ^AB,b^	834.19 ± 97.60 ^B,b^	517.46 ± 117.53 ^A,a^	682.04 ± 151.90 ^B,ab^	719.70 ± 199.05 ^B^
Heavy Shoulder	950.37 ± 245.12 ^B,b^	954.16 ± 327.80 ^B,b^	742.36 ± 281.50 ^B,ab^	718.79 ± 123.99 ^B,a^	841.42 ± 270.50 ^C^
Mean	828.10 ± 223.87 ^c^	759.54 ± 276.63 ^bc^	656.98 ± 231.38 ^ab^	586.18 ± 216.92 ^a^	707.70 ± 253.16

Note: Different capital letters indicate differences in same column; different lowercase letters indicate differences in same row for each ceramic type.

## Data Availability

Not applicable.

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
