# Peer review of "In-Vitro Investigation of Marginal Adaptation and Fracture Resistance of Resin Matrix Ceramic Endo-Crown Restorations"

_materials, 2023, doi:10.3390/ma16052059_

Round 1
Reviewer 1 Report
I would recommend to add in introduction following items:
To test the hypotheses, this research will employ following methods, conducted across few phases, seeking to answer the following questions:
Questions to be answered to prove the hypothesis or to determine null hypothesis.
Add please general and clinical recommendations, based on results of the study.
Author Response
Dear Reviewer, thank you for your comments and suggestions. We are pleased to answer your questions. We tried to improve our manuscript by following your suggestions.
I would recommend to add in introduction following items:
To test the hypotheses, this research will employ following methods, conducted across few phases, seeking to answer the following questions:
Questions to be answered to prove the hypothesis or to determine null hypothesis.
Response: Thank you for your advice. Hypothesis and answers were added to introduction and discussion part.
The first null hypothesis of this study was that different margin designs will not affect the fracture resistance and marginal adaptation of endo-crown restorations. The second null hypothesis was that the type of current RMC materials will not affect the fracture resistance and marginal adaptation of endo-crown restorations.
The first null hypothesis of the study, that different margin designs will not affect the fracture resistance, was rejected. However, marginal adaptation of endo-crown restorations was affected, thereby confirming the hypothesis. The second null hypothesis, that the type of current RMC materials will not affect the fracture resistance and marginal adaptation of endo-crown restorations, was rejected completely.
Add please general and clinical recommendations, based on results of the study.
Response: Clinical recommendation was added to conclusion section. ‘As a margin configuration, it is recommended to use the heavy shoulder finish line type as an ideal margin type, and the butt-joint margin type should be avoided in order to ensure the fracture strength of the restoration.’ Thank you for your advice.
Reviewer 2 Report
Abstract:
The abstract needs to be completely revised. Verbal tense is inadequate, results and conclusion are lacking.
Introduction:
The introduction is too long. The text needs to be more objective.
Materials and methods:
The omnicam camera is a camera for intraoral scanning. Did the authors mean that a scan was performed outside the mouth?
Description of marginal adaptation analysis methodology and image needs improvement.
Discussion
The first paragraphs of the discussion address various papers analyzing molars, but the study analyzes premolars.
The first five paragraphs of the discussion do not discuss the results obtained in the study.
Check and correct information about thermal cycling:
Abstract: “All samples will be subjected to thermal cycling”
Discussion: “In the present study, the absence of the thermal cycle and chewing cycle stages can be considered as a limitation of the study, and further studies in which thermal cycle and chewing cycle are also investigated and compared are needed.”
Author Response
Responses to Reviewer #2
Dear Reviewer, thank you for your comments and suggestions. We are pleased to answer your questions. We tried to improve our manuscript by following your suggestions.
Abstract:
The abstract needs to be completely revised. Verbal tense is inadequate, results and conclusion are lacking.
Response: The abstract is fully revised and results and conclusion are added. Thank you.
Introduction:
The introduction is too long. The text needs to be more objective.
Response: The introduction fully revised. Thank you.
Materials and methods:
The omnicam camera is a camera for intraoral scanning. Did the authors mean that a scan was performed outside the mouth?
Response: Thank you for your comment. We used the extraoral scanner and the sentence was corrected ‘The 3D scanning process of the main model with 3 different finish line prepara-tions was performed with an extraoral scanner (Sirona inEOS X5, Dentsply Sirona, York, PA USA)’
Description of marginal adaptation analysis methodology and image needs improvement.
Response: We modified the description of marginal adaptation methodology and figure following your suggestions.
Discussion
The first paragraphs of the discussion address various papers analyzing molars, but the study analyzes premolars.
Response: We revised the discussion section according to your suggestion, thank you.
The first five paragraphs of the discussion do not discuss the results obtained in the study.
Response: We revised the discussion section according to your suggestion, thank you.
Check and correct information about thermal cycling:
Abstract: “All samples will be subjected to thermal cycling”
Discussion: “In the present study, the absence of the thermal cycle and chewing cycle stages can be considered as a limitation of the study, and further studies in which thermal cycle and chewing cycle are also investigated and compared are needed.”
Response: We checked the information about thermal cycling. The sentence “All samples will be subjected to thermal cycling” was removed from abstract.
Reviewer 3 Report
The present study investigated the marginal adaptation and fracture strength of CAD/CAM hybrid crowns. The study protocol is adequate and was conducted without major issues. However, there are some substantial problems in this manuscript and study.
First of all, although the manuscript was written in understandable English, an extensive native check is needed, and it should be done by a person who also knows dental research.
Second, the hybrid materials they chose are not very popular in some or maybe many countries. The authors failed to explain the reasons for selecting those materials or the structural and compositional differences between them. The researchers and clinicians may want to know the mechanism and science behind the results. The lack of that information placed this study on a consumer report level. If they simply wanted to compare the effect of the preparation designs, they did not need to use four hybrid materials without using any glass ceramic as a control.
Third, the value of testing the fracture strength by applying static loading without thermal or chewing cycles is questionable since it does not simulate clinical conditions. I believe the ISO standard requires repetitive loading for the fracture strength evaluation.
The authors must present their unique findings and analysis that overcome those shortcomings for publication.
Here are some areas the authors could review. In terms of English, more areas should be reviewed by a native checker.
Introduction
Treatment options for the restoration of endodontic treated teeth with high crown
damage are increasing
What is high crown damage?
The bonding surface of the endocrown restoration in devital teeth is higher than in
vital teeth [5].
What do the authors mean by the surface is higher?
With the development of adhesive technology, today's ceramics are strengthened,
acid-etched, and bonded to the tooth tissue with strong resin
tooth structure
the fact that the light source cannot fully reach the walls of the cavity chamber and the polymerization of the resin cement is not fully achieved.
I know the authors talk about curing light, but you must clarify the light source in the sentence.
In addition, since all of the failures in the studies were due to adhesion failure, they were reported to be repairable and new restorative materials with better properties were introduced [9].
Better than what? Glass ceramic or their old versions?
Therefore, this method is only indicated for in-vitro studies. Its important advantages are that it is a complex, noninvasive, inexpensive measurement method and has a low risk of failure [17].
Is being complex an advantage?
Materials and methods
Then, 3 main models were subjected to 3 separate finish line preparations as butt-joint, heavy chamfer and heavy shoulder.
Separate - Different
Could the authors describe the dimension of the preparations? How deep was the shoulder? 1.0mm? Was that a rounded shoulder? They may want to present the diagram of the finish lines.
The 3D scanning process of the main model with 3 different finish line preparations
was performed with an extraoral scanner (Sirona CEREC Omnicam, Dentsply Sirona, PA),
What version of CEREC software?
Table 1
Could the authors present the composition of each material?
The pulp chambers of the replicas were etched for cementation
Did the authors use phosphoric acid for etching? If the authors applied only Single Bond Universal, they might want to say just “Single Bond Universal was applied” to avoid confusion.
Results
The mean values of marginal adaptation for all groups are depicted in Table 1.
Table 2?
While the highest microleakage was observed in Grandio (36.53±12.01),
The authors did not test microleakage but the marginal gap. The same goes for Table 2 (should be 3).
Table 2
Total sounds more like the summation of values. Would the authors like to change it to “average” or “mean”?
The mean values and standard deviations of fracture resistance for different CAD-CAM blocks of endocrown restorations are presented in Table 2.
Table 3?
Discussion
In a study comparing endocrown with conventional crown restorations, it was reported that endocrowns exhibited better fracture resistance [29].
Is that a crown fracture or a tooth fracture?
Attia et al. found no significant difference in the average fracture resistance of composite resin and all-ce- ramic CAD/CAM crowns in their study [33], while another study reported a better result when using composite restorations [34].
I know the authors investigated fracture strength, but how about the wear resistance? The resistance to fracture is not the only property required for endocrowns. Any significant difference between them in terms of wear resistance in the previous study?
region in the patient's mouth were in the range of 200-560 N [36]. Hassouneh et al. concluded that endochrons obtained from CAD-CAM resin composite blocks are a reliable option for restoring endodontically treated premolars [7].
endocrowns
Author Response
Responses to Reviewer #3
The present study investigated the marginal adaptation and fracture strength of CAD/CAM hybrid crowns. The study protocol is adequate and was conducted without major issues. However, there are some substantial problems in this manuscript and study.
Dear Reviewer, thank you for your comments and suggestions. We are pleased to answer your questions. We tried to improve our manuscript by following your suggestions.
First of all, although the manuscript was written in understandable English, an extensive native check is needed, and it should be done by a person who also knows dental research.
Response: The English language of our publication was edited by a dental researcher and by the journal's Language Editing Services, through your suggestions. Thank you.
Second, the hybrid materials they chose are not very popular in some or maybe many countries. The authors failed to explain the reasons for selecting those materials or the structural and compositional differences between them. The researchers and clinicians may want to know the mechanism and science behind the results. The lack of that information placed this study on a consumer report level. If they simply wanted to compare the effect of the preparation designs, they did not need to use four hybrid materials without using any glass ceramic as a control.
Response: Thank you for warning us about this. With your valuable comments, we have added why we preferred RMC materials in this study to the introduction section as follows.
‘Recently introduced resin matrix ceramics (RMCs) consist of materials with a highly infiltrated organic matrix with ceramic particles and have a modulus of elasticity closer to dentin than conventional ceramics. RMCs merge the advantageous characteristics of dental ceramics and composite resins and thereby provide the following benefits: easy and fast machinability, can be repaired intraorally, less chipping, better fracture resistance, acceptable wear resistance, low abrasive effect on opposing teeth, near-to-ideal bond strength, polishability and no need for firing [5,6,7,8].’
Third, the value of testing the fracture strength by applying static loading without thermal or chewing cycles is questionable since it does not simulate clinical conditions. I believe the ISO standard requires repetitive loading for the fracture strength evaluation.
Response: Thank you for your suggestion. We are in agreement with you that the evaluation of these parameters would improve our study. However, we evaluated the parameters within the range of our funding and these properties may be investigated in further studies. In addition, thermal cycle and chewing cycle parameters have been added to the limitation part of the publication for the reason we have explained to you.
The authors must present their unique findings and analysis that overcome those shortcomings for publication.
Here are some areas the authors could review. In terms of English, more areas should be reviewed by a native checker.
Introduction
Treatment options for the restoration of endodontic treated teeth with high crown
damage are increasing
What is high crown damage?
Response: It was changed to ‘extensive crown damage’. Thank you.
The bonding surface of the endocrown restoration in devital teeth is higher than in
vital teeth [5].
What do the authors mean by the surface is higher?
Response: The introduction section fully revised. We shortened the introduction. Thank you.
With the development of adhesive technology, today's ceramics are strengthened,
acid-etched, and bonded to the tooth tissue with strong resin
tooth structure
Response: The introduction section fully revised. We shortened the introduction. Thank you.
the fact that the light source cannot fully reach the walls of the cavity chamber and the polymerization of the resin cement is not fully achieved.
I know the authors talk about curing light, but you must clarify the light source in the sentence.
Response: Thank you for your advice. In material method part the sentence revised to ‘A halogen light source’
In addition, since all of the failures in the studies were due to adhesion failure, they were reported to be repairable and new restorative materials with better properties were introduced [9].
Better than what? Glass ceramic or their old versions?
Response: The introduction section fully revised. We shortened the introduction. Thank you.
Therefore, this method is only indicated for in-vitro studies. Its important advantages are that it is a complex, noninvasive, inexpensive measurement method and has a low risk of failure [17].
Is being complex an advantage?
Response: The sentence was fully revised in discussion part.
‘This technique is a non-invasive, inexpensive procedure that allows accurate, reproducible measurements to be made at various points, allowing both marginal and in-ternal adaptation of restorations to be examined’
Materials and methods
Then, 3 main models were subjected to 3 separate finish line preparations as butt-joint, heavy chamfer and heavy shoulder
Separate – Different
Response: It was changed to “different”. Thank you.
Could the authors describe the dimension of the preparations? How deep was the shoulder? 1.0mm? Was that a rounded shoulder? They may want to present the diagram of the finish lines.
Response: Thank you for your advice. The margin design of the heavy chamfer and heavy shoulder were prepared to be equal all around and 1 mm wide.
The 3D scanning process of the main model with 3 different finish line preparations
was performed with an extraoral scanner (Sirona CEREC Omnicam, Dentsply Sirona, PA),
What version of CEREC software?
Response: Thank you for your comment. The sentence was corrected and the version of CAD program was added.
‘The 3D scanning process of the main model with 3 different finish line preparations was performed with an extraoral scanner (Sirona inEOS X5, Dentsply Sirona, York, PA USA), and the obtained digital data were transferred to computer-aided design (CAD) software (inLab SW 16.1; Dentsply Sirona).’
Table 1
Could the authors present the composition of each material?
Response: We modified the table 1 following your suggestions.
The pulp chambers of the replicas were etched for cementation
Did the authors use phosphoric acid for etching? If the authors applied only Single Bond Universal, they might want to say just “Single Bond Universal was applied” to avoid confusion.
Response: Thank you for your advice. Phosphoric acid for etching wasn’t used. And the sentence was changed ‘Single Bond Universal was applied to the pulp chambers of the replicas (Single Bond Universal, 3M ESPE, St. Paul, MN USA).’
Results
The mean values of marginal adaptation for all groups are depicted in Table 1.
Table 2?
Response: It was changed to “table 2”. Thank you.
While the highest microleakage was observed in Grandio (36.53±12.01),
The authors did not test microleakage but the marginal gap. The same goes for Table 2 (should be 3).
Response: It was changed to “highest marginal gap values”. Thank you.
Table 2
Total sounds more like the summation of values. Would the authors like to change it to “average” or “mean”?
Response: It was changed to “mean”. Thank you.
The mean values and standard deviations of fracture resistance for different CAD-CAM blocks of endocrown restorations are presented in Table 2.
Table 3?
Response: It was changed to “Table 3”. Thank you.
Discussion
In a study comparing endocrown with conventional crown restorations, it was reported that endocrowns exhibited better fracture resistance [29].
Is that a crown fracture or a tooth fracture?
Response: The sentence was revised.
‘In a study comparing endo-crowns with conventional crown restorations, it was reported that endo-crowns exhibited better tooth fracture resistance and also reported that the fracture resistance of endo-crowns was greater than that of intact premolars’ thank you.
Attia et al. found no significant difference in the average fracture resistance of composite resin and all-ceramic CAD/CAM crowns in their study [33], while another study reported a better result when using composite restorations [34].
I know the authors investigated fracture strength, but how about the wear resistance? The resistance to fracture is not the only property required for endocrowns. Any significant difference between them in terms of wear resistance in the previous study?
Response: Thank you for your suggestion we are in agreement with you that it is not enough to only look at the fracture resistance in order to use a material. however, since there are two parameters in our study. If one more parameter is added, it will be difficult to conclude the comments. In addition, we evaluated the parameters within the range of our funding and these properties may be investigated in further studies.
region in the patient's mouth were in the range of 200-560 N [36]. Hassouneh et al. concluded that endochrons obtained from CAD-CAM resin composite blocks are a reliable option for restoring endodontically treated premolars [7].
endocrowns
Response: It was changed to “endocrowns”. Thank you.

Round 2
Reviewer 2 Report
The manuscript has greatly improved.
In the methodology, I suggest standardizing the name of the Ambarino material (Ambarino High Class or Creamed Ambarino High Class Blanc)
Answer the hypotheses in the discussion, according to the results obtained.
Results: “It was observed that the type of preparation did not affect the marginal adaptation”
Discussion: The first null hypothesis of the study, that different margin designs will not affect the fracture resistance, was rejected. However, marginal adaptation of endo-crown restorations was affected, thereby confirming the hypothesis”
Author Response
Responses to Reviewer #2
The manuscript has greatly improved.
Dear Reviewer, thank you for your comments and suggestions. We are pleased to answer your questions. We tried to improve our manuscript by following your suggestions.
In the methodology, I suggest standardizing the name of the Ambarino material (Ambarino High Class or Creamed Ambarino High Class Blanc)
Response: Thank you. It was standardized to ‘Ambarino High Class’.
Answer the hypotheses in the discussion, according to the results obtained.
Results: “It was observed that the type of preparation did not affect the marginal adaptation”
Discussion: The first null hypothesis of the study, that different margin designs will not affect the fracture resistance, was rejected. However, marginal adaptation of endo-crown restorations was affected, thereby confirming the hypothesis”
Response: It was revised. ‘The first null hypothesis of the study, that different margin designs will not affect the fracture resistance, was rejected. However, marginal adaptation of endo-crown restorations was not affected, thereby confirming the hypothesis. Thank you.

Reviewer 3 Report
The authors extensively revised the manuscript in response to the reviewers’ comments, and the quality of the paper was significantly increased.
There are a few suggestions.
Could you add sentences that describe the reasons for the different fracture resistance values depending on those four materials? For example, what compositional differences made the fracture strength difference between the values of Voco Grnadio and Brilliant Crios, although they are both in the same category of materials? The manufacturers never disclose the complete information, but the authors may want to give readers assumptions.
“Attia et al. found no significant difference in the average fracture resistance of composite resin and all-ceramic CAD/CAM crowns in their study [33], while another study reported a better result when using composite restorations [34].
I know the authors investigated fracture strength, but how about the wear resistance? The resistance to fracture is not the only property required for endocrowns. Any significant difference between them in terms of wear resistance in the previous study?
Response: Thank you for your suggestion we are in agreement with you that it is not enough to only look at the fracture resistance in order to use a material. however, since there are two parameters in our study. If one more parameter is added, it will be difficult to conclude the comments. In addition, we evaluated the parameters within the range of our funding and these properties may be investigated in further studies.”
The reviewer did not ask the authors to add another experiment or parameter in this part. The author could comment on the “previous study” that compared wear resistance between hybrid and glass-ceramics or others. Because if the fracture resistance and marginal fit are sufficient, the next clinically important physical property would be wear resistance. It was not tested by the present study, which is fine or even better, but the authors should not conclude that hybrid material is the best for the endo-crown cases without mentioning that property. One sentence should be enough.
The reviewer should have pointed it out at the first review, but in the conclusion section, the sentence, “3. Since the fracture resistance of all materials is higher than the force values obtained during chewing in the premolar region, the use of these materials in the restoration of endodontically treated premolars is ideal.”may need to be removed or revised. We should not simply compare the mastication force in the oral cavity and the experimental static load. Also, since the authors did not test the other properties, such as wear resistance, the word “ideal” sounds too definitive.
Author Response
Responses to Reviewer #3
The authors extensively revised the manuscript in response to the reviewers’ comments, and the quality of the paper was significantly increased.
Dear Reviewer, thank you for your comments and suggestions. We are pleased to answer your questions. We tried to improve our manuscript by following your suggestions.
There are a few suggestions.
Could you add sentences that describe the reasons for the different fracture resistance values depending on those four materials? For example, what compositional differences made the fracture strength difference between the values of Voco Grandio and Brilliant Crios, although they are both in the same category of materials? The manufacturers never disclose the complete information, but the authors may want to give readers assumptions.
Response: Thank you for your suggestions. We added sentences that describe the reasons for the different fracture resistance values at discussion part.
“Attia et al. found no significant difference in the average fracture resistance of composite resin and all-ceramic CAD/CAM crowns in their study [33], while another study reported a better result when using composite restorations [34].
I know the authors investigated fracture strength, but how about the wear resistance? The resistance to fracture is not the only property required for endocrowns. Any significant difference between them in terms of wear resistance in the previous study?
Response: Thank you for your suggestion we are in agreement with you that it is not enough to only look at the fracture resistance in order to use a material. however, since there are two parameters in our study. If one more parameter is added, it will be difficult to conclude the comments. In addition, we evaluated the parameters within the range of our funding and these properties may be investigated in further studies.”
The reviewer did not ask the authors to add another experiment or parameter in this part. The author could comment on the “previous study” that compared wear resistance between hybrid and glass-ceramics or others. Because if the fracture resistance and marginal fit are sufficient, the next clinically important physical property would be wear resistance. It was not tested by the present study, which is fine or even better, but the authors should not conclude that hybrid material is the best for the endo-crown cases without mentioning that property. One sentence should be enough.
Response: Thank you for your suggestions. We added a sentence to discussion part.
The reviewer should have pointed it out at the first review, but in the conclusion section, the sentence, “3. Since the fracture resistance of all materials is higher than the force values obtained during chewing in the premolar region, the use of these materials in the restoration of endodontically treated premolars is ideal.”may need to be removed or revised. We should not simply compare the mastication force in the oral cavity and the experimental static load. Also, since the authors did not test the other properties, such as wear resistance, the word “ideal” sounds too definitive.
Response: Thank you for your suggestions. In line with what you said, we removed the relevant sentence from the result section and added it to the discussion section by revising it.
